# PROVE—Pre-Eclampsia Obstetric Adverse Events: Establishment of a Biobank and Database for Pre-Eclampsia

**DOI:** 10.3390/cells10040959

**Published:** 2021-04-20

**Authors:** Lina Bergman, Karl Bergman, Eduard Langenegger, Ashley Moodley, Stephanie Griffith-Richards, Johan Wikström, David Hall, Lloyd Joubert, Philip Herbst, Sonja Schell, Teelkien van Veen, Michael Belfort, Stephen Y. C. Tong, Susan Walker, Roxanne Hastie, Catherine Cluver

**Affiliations:** 1Department of Women’s and Children’s Health, Uppsala University, 75309 Uppsala, Sweden; 2Department of Obstetrics and Gynaecology, Institute of Clinical Sciences, Sahlgrenska Academy, University of Gothenburg, 41650 Gothenburg, Sweden; 3Department of Obstetrics and Gynaecology, Faculty of Medicine and Health Sciences, Stellenbosch University, Cape Town 7505, South Africa; langen@sun.ac.za (E.L.); ashleymoodley69@gmail.com (A.M.); drh@sun.ac.za (D.H.); sonjaschell@sun.ac.za (S.S.); cathycluver@hotmail.com (C.C.); 4Department of Molecular and Clinical Medicine, Institution of Medicine, Sahlgrenska Academy, University of Gothenburg, 41650 Gothenburg, Sweden; karl.bergman@gu.se; 5Division of Radiodiagnosis, Department of Medical Imaging and Clinical Oncology, Faculty of Medicine and Health Sciences, Stellenbosch University, Cape Town 7505, South Africa; drsteph21@hotmail.com; 6Department of Surgical Sciences, Radiology, Uppsala University, 75309 Uppsala, Sweden; johan.wikstrom@radiol.uu.se; 7Division of Cardiology, Department of Medicine, Medicine and Health Sciences, Stellenbosch University, Cape Town 7505, South Africa; lloydjoubert@gmail.com; 8Department of Obstetrics and Gynaecology, University Medical Center Groningen, 9713 Groningen, The Netherlands; teelkien@gmail.com (T.v.V.); herbst@hotmail.com (P.H.); 9Department of Obstetrics and Gynaecology, Baylor College of Medicine, Houston, TX 77004, USA; belfort@bcm.edu; 10Translational Obstetrics Group, Department of Obstetrics and Gynaecology, University of Melbourne, Parkville, VIC 3010, Australia; stong@unimelb.edu.au (S.Y.C.T.); spwalker@unimelb.edu.au (S.W.); hastie.r@unimelb.edu.au (R.H.); 11Mercy Perinatal, Mercy Hospital for Women, Heidelberg, VIC 3010, Australia

**Keywords:** pre-eclampsia, eclampsia, pulmonary oedema, biobank, database

## Abstract

Pre-eclampsia is a leading cause of maternal and perinatal morbidity and mortality. The burden of disease lies mainly in low-middle income countries. The aim of this project is to establish a pre-eclampsia biobank in South Africa to facilitate research in the field of pre-eclampsia with a focus on phenotyping severe disease.The approach of our biobank is to collect biological specimens, detailed clinical data, tests, and biophysical examinations, including magnetic resonance imaging (MRI) of the brain, MRI of the heart, transcranial Doppler, echocardiography, and cognitive function tests.Women diagnosed with pre-eclampsia and normotensive controls are enrolled in the biobank at admission to Tygerberg University Hospital (Cape Town, South Africa). Biological samples and clinical data are collected at inclusion/delivery and during the hospital stay. Special investigations as per above are performed in a subset of women. After two months, women are followed up by telephonic interviews. This project aims to establish a biobank and database for severe organ complications of pre-eclampsia in a low-middle income country where the incidence of pre-eclampsia with organ complications is high. The study integrates different methods to investigate pre-eclampsia, focusing on improved understanding of pathophysiology, prediction of organ complications, and potentially future drug evaluation and discovery.

## 1. Introduction

Pre-eclampsia is one of the most serious complications of pregnancy and affects 3–8% of pregnancies worldwide [1]. It is a leading cause of maternal and perinatal morbidity and mortality [2]. Globally, pre-eclampsia is responsible for more than 60,000 maternal deaths annually, and in South Africa, hypertensive disorders of pregnancy account for 14% of all maternal deaths [3].

The pathogenesis of pre-eclampsia is not yet fully elucidated. It likely involves maternal, foetal, and placental factors, resulting in relative placental under-perfusion, hypoxia, and ischaemia [1]. The placenta then releases anti-angiogenic factors causing endothelial dysfunction resulting in a multi-system disorder [4]. Hypertension is the hallmark feature and other systems including renal, cardiovascular, respiratory, haematological, hepatic, cerebrovascular, and the fetoplacental unit can be involved [5]. In its most severe form, it can cause seizures (eclampsia), cerebrovascular events, pulmonary oedema, cardiac failure, acute kidney injury, haemolysis, elevated liver enzymes, low platelet (HELLP) syndrome, disseminated intravascular coagulation, sub-capsular liver haematoma or rupture, and even death [6,7,8,9,10].

Pre-eclampsia is a disease that is only observed in humans. No animal models of pre-eclampsia have the ability to exactly mimic the true pre-eclampsia state due to differences in placentation among mammals [11]. It is imperative that critical laboratory observations are made on human tissues.

Biobanks enable research for which clinical information and analyses from biological samples are efficiently used and combined in more than one project. Biobanks can also be used in collaboration with international biobanks to achieve sufficient power and generalisability in rare outcomes.

Low-and-middle-income countries (LMIC) have a higher incidence of pre-eclampsia, more complications, and bear the major burden of global maternal and perinatal deaths [6]. For these reasons, it makes sense to establish a biobank in an area with a high incidence of pre-eclampsia. Researchers from Tygerberg Hospital, Stellenbosch University, have been investigating pre-eclampsia for many years [12,13,14,15,16]. Tygerberg hospital is a referral center servicing a population of over 2 million people. The Obstetrics Department only manages complicated pregnancies (such as those affected by severe, or preterm pre-eclampsia) and delivers approximately 8000 women per year. The high-risk referral service is supported by maternal-foetal medicine sub-specialists, critical care specialists, and neonatologists. The occurrence of pre-eclampsia and its complications is high in South Africa which is expected for a low-and-middle-income country. Unfortunately, the exact incidence is not known. The academic centre at Tygerberg Hospital is, therefore, an ideal service to embed a pre-eclampsia biobank.

The research team at Tygerberg Hospital also has established collaborations with leading academic centres in Australia, Sweden, The Netherlands, the United Kingdom, and the United States of America. Such collaborations enable linkage of a unit that has an extremely high incidence of pre-eclampsia, and in particular, pre-eclampsia with severe features, with other research leaders in the field of pre-eclampsia. These collaborations can enable us to make important discoveries about why different organs are affected in pre-eclampsia, findings that may improve the clinical care of women.

## 2. Aims and Objectives

To establish a pre-eclampsia database and biobank at Tygerberg Hospital, Stellenbosch University and to facilitate research in pre-eclampsia, detailed data are collected according to an international consensus of core predictors and outcomes in pre-eclampsia research [17]. The aims of this approach are to enable the discovery of possible predictors and diagnostic tools for organ complications in pre-eclampsia and to develop the understanding of the underlying pathophysiology further with a focus on different organ complications. We collect biological specimens including plasma, saliva, cerebrospinal fluid, urine, placenta, and umbilical cord blood (mixed venous–arterial). In subgroups, we perform specialised examinations to assess organ dysfunction, reveal underlying pathophysiological pathways, and aim to relate the findings to predictors and biomarkers. These examinations include cerebral magnetic resonance imaging (MRI), cardiac MRI, transcranial Doppler, echocardiography, and cognitive function testing.

## 3. Materials and Methods

The study is registered at the International Standard Randomised Controlled Trial Number (ISCRTN) with trial registration number ISRCTN10623443.

### 3.1. Study Design and Population

We include women who are admitted to Tygerberg Hospital for the birth of their baby. Cases (women with pre-eclampsia) are included before or after delivery depending on presentation. Controls (women with normotensive pregnancies) are recruited when admitted for delivery or at a visit to the outpatient clinic. For the controls, samples are collected at inclusion and delivery. Cases are seen daily and clinical information and samples are collected serially until discharge. All participants are contacted telephonically in the postpartum period. Data are collected prospectively and from the medical charts on maternal health and neonatal outcomes (Figure 1).

#### Inclusion and Exclusion Criteria

All women with pre-eclampsia who are admitted to Tygerberg hospital are eligible for inclusion. Our controls are women who deliver at Tygerberg without the hypertensive disease. Pre-eclampsia is defined according to the International Society for the Study of Hypertension in Pregnancy (ISSHP) [18]. Both women with pre-eclampsia without severe features and pre-eclampsia with end-organ complications are eligible for inclusion. End-organ complications are defined according to the ISSHP classification system and include haemolysis, elevated liver enzymes, and low platelets (HELLP) syndrome, eclampsia, pulmonary oedema, renal impairment, and acute severe hypertension [18]. All parameters are included in the online database described below, and women can thus be classified according to different classification systems. All participants must be competent to provide informed consent before enrolment. If they are minors, they need to read and sign the assent form, and a parent or guardian must sign consent. All individual studies originating from the biobank require a unique ethical approval.

### 3.2. Collection of Clinical Information

Clinical information is collected on datasheets and entered into an online secure password-protected Research Electronic Data Capture (REDCap) database. The data collection includes information on the participant’s age, gravidity, parity, medical history, current pregnancy information, and outcome of the pregnancy. Only the principal investigators have access to patient identifying information and each participant is allocated a study number. The variables collected in the database correspond to the recommended core demographics, predictors, and outcomes listed in “*Strategy for standardization for pre-eclampsia research design*” by the international group Co-Lab for harmonisation of pre-eclampsia research (Table 1) [17]. The women are also interviewed at inclusion with a short questionnaire regarding signs and symptoms specific for pre-eclampsia (Table 1).

### 3.3. Sample Collection and Storage

Samples are collected by research midwives. After a blood sample is drawn, it is centrifuged, aliquoted, and frozen within two hours of collection. The same procedure is performed for cerebrospinal fluid and umbilical cord blood. Saliva and urine are frozen directly and are not centrifuged. Placental samples are taken at four sites on the maternal and foetal surfaces of the placenta. They are stored in RNAlater for at least 48 h and then frozen. All samples are stored in a minus 70 °C freezer which has an electronic temperature monitoring system. All samples are labelled only with the study number, sample number, and date. Samples collected for the biobank are presented in Table 2.

### 3.4. Questionnaires and Biophysical Examinations

#### 3.4.1. Cognitive Function Testing

Cognitive function is assessed subjectively and objectively as close to discharge as possible. The subjective function is assessed using the Cognitive Failure Questionnaire (Appendix A). Cognitive function is also assessed objectively using the Montreal Cognitive Assessment (MoCA) (Appendix A). A detailed description of the cognitive function tests can be found in the Appendix A.

#### 3.4.2. Brain MRI

For women with eclampsia and women with pre-eclampsia with severe features without eclampsia as a control group, we perform MRI examinations on entry into the study. We use a 1.5 Tesla scanner, used in clinical practice, at the Department of Radiology at Tygerberg University Hospital. The MRI protocol includes sequences for evaluation of brain morphology including infarcts, oedema, and haemorrhages, as well as assessment of arterial spasm and cerebral perfusion. Imaging sequence information can be found in the Appendix A.

#### 3.4.3. Transcranial Doppler

Transcranial Doppler examination is performed in a subgroup of women to evaluate the cerebral perfusion pressure and dynamic cerebral autoregulation on entry into the study and for some women also before discharge from the hospital and in addition before and after vasoactive medication. Both women with normotensive pregnancies, pre-eclampsia without severe features, and pre-eclampsia with severe features are eligible for transcranial Doppler examination. This is performed by locating the middle cerebral artery bilaterally by the Doppler technique and simultaneously recording continuous blood pressure and end-tidal CO_2_. A detailed description of the methodology can be found in the Appendix A.

#### 3.4.4. Echocardiography and Cardiac MRI

A subgroup of cases (women with pulmonary oedema) and controls (women with pre-eclampsia without pulmonary oedema and women with normotensive pregnancies) have echocardiograms and cardiac MRI on entry into the study. Investigations are performed by two cardiologists according to a predefined protocol that can be found in the Appendix A.

### 3.5. Statistics

Results from echocardiography, MRI, and cerebral Doppler will be calculated by two independent interpreters blinded to groups and entered manually into the database. MR data will be analysed in collaboration with Uppsala University by neuroradiologists, blinded to groups. Blood samples will be analysed for placental, cardiac-, renal-, neurological- and endothelial biomarkers, using standardised platforms when available and for the remaining analyses through manual enzyme-linked immunosorbent assay (ELISA) analyses in duplicates. Inter- and intra-assay coefficients of variation will be aimed at below 10%.

Demographics will be presented as medians or means as appropriate by distribution. Comparison between groups will be analysed by Student’s *t*-test or Mann–Whitney u-test with means or medians and confidence intervals or interquartile range, as appropriate according to the distribution of the variables. When comparing multiple groups, the Kruskal–Wallis test or one-way ANOVA will be used as appropriate according to the distribution of the variables. Correlations will be analysed by Pearson’s r or Spearman’s rho, as appropriate by the distribution of the variable. Regression analyses, unadjusted and adjusted, will be performed to adjust for known confounding variables. A total of 10 cases per variable at the lowest will be considered appropriate to avoid overfitting of the model. All statistical analyses will be performed in SPSS or R.

#### Power Calculations

Tygerberg University hospital has approximately 8000 high-risk deliveries yearly. Pre-eclampsia affects a large proportion of these deliveries but the exact numbers are not known.

Prospective power calculations for some investigations will not be possible since no reference values or results exist. In addition, the biobank is designed to be open for future research on pre-eclampsia where the research question to date might not be known. Some examples of power calculations for planned analyses are given below.

Cerebral blood flow regulation: To detect a difference in dynamic cerebral autoregulation index of 1, 2 with a standard deviation (SD) of 1.5, 25 women are required in each group [19]. Cerebral biomarkers: To detect a difference of 4 pg/mL between cases and controls for NfL with an SD of 5, 25 women are required per group [20].

In order to extend the comparisons to correlations and sub-analyses, the initial sample size for PROVE was set to 100 women with eclampsia, 50 women with pulmonary oedema, and 50 women in each control group before the first round of analyses are initiated.

### 3.6. Patient and Public Involvement

No input from patients has been solicited in the creation of the database and biobank.

### 3.7. Ethics and Dissemination

Due to the frequency, morbidity, and mortality associated with pre-eclampsia, there is a great scientific and social value to undergird this research. Blanket and broad consent have been avoided, and only research linked to the topic of pre-eclampsia will be conducted. The risk of adverse events is very low. Peripheral blood collection is carried out by an experienced midwife/doctor at the time of routine blood collection. Cerebrospinal fluid is collected from a discarded sample from the anaesthetist during the administration of spinal anaesthesia. MRI imaging in selected participants with eclampsia or cerebral signs has fewer side effects than conventional imaging. Performing transcranial Doppler measurements holds no risks. Participant information databases will be routinely backed up to prevent loss due to technical issues, and samples within the biobank will be secured via routine laboratory safety nets (e.g., the availability of backup freezer space, documentation of usage, etc.).

Blood samples are collected at the same time as routine blood testing, and thus, no additional discomfort is inflicted. Other samples are collected from tissue and fluid that are usually discarded.

No publications will come directly from the biobank. The biobank will provide a source of material for further studies that will each need individual ethics approval.

## 4. Collection to Date

From April 2018 until March 2020, 244 women have been included in the biobank. The biobank is an ongoing project that will continuously enrol women with pre-eclampsia and controls with yearly updates to the local Health Research Ethics Committee. In Table 3, the background data of these women are presented according to the type of organ complication at inclusion. The biobank recently restarted inclusions after the COVID-19 pandemic in March 2021 and will continue to include women according to the research questions and sample size described above.

## 5. Discussion

This project aims to establish a biobank and database for severe organ complications of pre-eclampsia. This is only feasible in LMIC where the incidence of pre-eclampsia and its complications are high. To our knowledge, the project is the first biobank of its kind in an LMIC, integrating different methods and engaging clinical and preclinical researchers to investigate the organ complications of pre-eclampsia. The project has a unique focus on improving the understanding of the pathophysiology, the possibility for prediction of organ complications, and over time, the hope for drug evaluation and drug discovery for pre-eclampsia with severe features.

This particular field of research is hampered by small retrospective studies that lack sufficient validation or power [21]. Organ complications of pre-eclampsia are complex and difficult to investigate due to their low incidence in high-income countries, their sudden onset, and the restriction often imposed by research methods in pregnant women. By using a database that adheres to recommended variables and biosamples in pre-eclampsia research, the results from this study can be merged with results from other studies. This will enable the investigation of infrequent rare outcomes by increasing sample size, this providing valuable scientific knowledge, and support to the women who are affected by this serious condition [17]. PROVE holds biological samples and data from special investigations such as MRI and cerebral blood flow measurements as well as detailed demographic data. Taken together, this will enable us to examine the disease phenotype deeply and achieve an improved understanding of the pathophysiology behind the various debilitating organ complications.

Another known biobank of pregnancy complication in an LMIC is PREPARE [22]. PREPARE collects biospecimens and clinical data from women with pre-eclampsia in Brazil and also uses the same recommended variables as PROVE does from the CoLab initiative [17]. The aim of PREPARE is to include a large sample size of women in order to validate existing biomarkers and diagnostic tools in an LMIC setting, whereas in PROVE, we have a smaller sample size, but we also perform special investigations, focusing on underlying pathophysiological mechanisms. These biobanks are complementary to each other and also enable the merging of data and biomarker results. Other examples of existing biobanks (mostly in HIC) are the Baby Biobank [23], Peribank [24], and IMPACT study [25] that focus on the prediction of disease, genetics, and microbiome.

Studies assessing neurological complications of pre-eclampsia are mostly small retrospective case-control and preclinical studies investigating cerebral blood flow alterations, cerebral biomarkers, MRI findings, and clinical signs and symptoms. Results indicate altered dynamic cerebral autoregulation and increased peripheral concentrations of cerebral biomarkers in women with pre-eclampsia, compared to normotensive women [19,20,26]. However, these results are from smaller populations and have not been investigated in women with clinical manifestations of cerebral complications such as eclampsia. In PROVE, we add knowledge to this field by adding similar investigations in a group of women with severe disease. Again, this should improve the understanding of the pathophysiology underlying cerebral oedema, eclampsia, and intracerebral haemorrhage in pre-eclampsia. Cerebral biomarkers and cognitive function deficits in previous reports have also mainly been based on women with less severe disease [20,26,27]. Results from PROVE will confirm or dismiss the role of cerebral biomarkers and cognitive function in relation to the severity of disease in a dose–response fashion. Clinical signs and symptoms are poorly investigated and in a recent systematic review by our group, we showed that clinical signs and symptoms currently used such as headache and visual disturbances are generally poor predictors of eclampsia, and included studies were mainly composed of case–control studies [28]. Our study collects signs and symptoms in a predefined protocol, enabling the discovery of new promising predictors of eclampsia, as opposed to predictors already collected in clinical practice. These data will be combined with data collected in a similar fashion in other countries such as Pakistan, Solomon Islands, and Sweden. MRI findings are predominantly retrospective studies in which the prevalence of cerebral oedema in eclampsia varies from 60 to 100%, and MRI has been performed mostly for by clinical indications. The prevalence of cerebral oedema in other severe forms of pre-eclampsia is still unknown [29,30]. Thus, there is a need for larger, studies incorporating prospectively collected information with predefined variables and systematically conducted investigations of the cerebral function in women with pre-eclampsia, with and without neurological complications.

Pulmonary oedema is characterised by fluid retention in the alveoli causing swelling of the lungs. Fluid is normally kept within the capillaries due to healthy endothelial cells and a balance between the opposing capillary hydrostatic pressure and colloid oncotic pressure. If this balance is deranged or the endothelial layer in the alveoli is disrupted, pulmonary oedema can follow [31]. Pulmonary oedema secondary to pre-eclampsia can theoretically be caused by any of these pathways, explained by increased hydrostatic pressures due to increased afterload and left ventricular diastolic dysfunction in combination with capillary leak and interstitial oedema [32,33,34]. There is a paucity of data regarding the pathophysiological pathways in pulmonary oedema and how to individualise treatment depending on the underlying causes such as heart failure treatment versus lowering of blood pressure and fluid restriction.

The diagnosis of pulmonary oedema in pre-eclampsia is based on clinical and radiological features, and echocardiography is not a prerequisite for diagnosis [31]. Thus, the contribution of the heart in pre-eclampsia complicated by pulmonary oedema remains largely unknown. There is, therefore, a need to evaluate diagnostic cardiac biomarkers and/or echocardiography in the management of women with pre-eclampsia and pulmonary oedema regarding the involvement of the heart to improve short- and long-term prognosis. Data from PROVE will contribute to this gap in knowledge by a deeper characterisation of cardiac tissue abnormalities, cardiac function, and cardiac biomarkers in women with pulmonary edema versus women with pre-eclampsia that is not complicated by pulmonary oedema and normotensive pregnancies, respectively.

In summary, PROVE aims to improve knowledge of the pathophysiology behind the development of severe organ complications of pre-eclampsia, enabling early identification and treatment and in addition, over time, identification of new drug targets. PROVE is open to all researchers after application to the research team and ethical approval. More information can be found at www.preeclampsiaresearch.com (accessed on 17 April 2021). Through the high-quality standardised collection of data and biosamples in accordance with the requirements of CoLab [17], PROVE will contribute to data- and biosample sharing in infrequent outcomes, enabling analyses of larger datasets. This is urgently needed in order to contribute to the sustainable development goals and improve women’s health.

## Figures and Tables

**Figure 1 cells-10-00959-f001:**
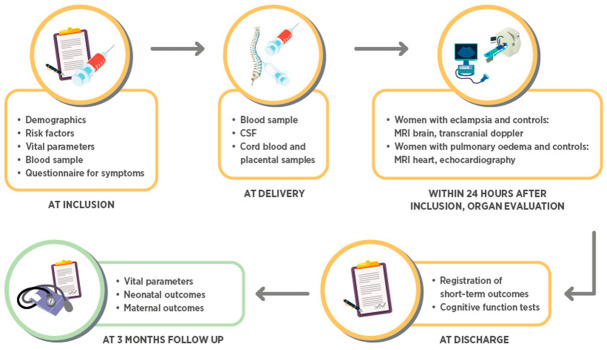
Flow chart of the study.

**Table 1 cells-10-00959-t001:** Clinical information collected in the database RedCap.

Variables at Inclusion	Variables During Hospital Stay	Variables at Discharge and at 3 Months Follow Up
**Demographics** Age (c)	**Delivery details** Gestation at delivery (c)	**Discharge** Date of discharge (N/A)
Date of birth (N/A)	Indication for delivery (n)	Number of days in hospital (c)
Hospital folder number (N/A)	Mode of delivery (n)	Adverse maternal outcomes (n)
Address (N/A)	Place and supervision of delivery (n)	*Eclampsia*
Contact number (N/A)	Medication used before delivery (n)	*Recurrent eclampsia*
Race (n)	Maternal plasma sample taken (b)	*Severe hypertension*
Marital status (n)	Cerebrospinal fluid sample taken (b)	*Stroke*
Total years of full-time education (o)		*PRES syndrome*
Current job situation (n)	**Neonatal outcomes**	*Cortical blindness*
Current living situation (n)	Liveborn (b)	*Severe renal impairment*
Number of people cohabitating (o)	Sex (b)	*Dialysis*
Diagnosis at inclusion (n)	Intubation at birth (b)	*Pulmonary edema*
Singleton pregnancy (b) Plasma sample taken at inclusion (b)	Birth weight (c) APGAR score at 5 min (o)	*Inotropic support* *Signs of bleeding or DIC*
**Medical history** Gravidity (o) Parity (o) Number of miscarriages (o) Number of previous terminations (o) Outcome of previous viable pregnancies (n) Previous pre-eclampsia, number of times (o) Previous pre-eclampsia before 34 weeks (b) New paternity (b) Fertility treatment (n) HIV status (b) If HIV positive, viral load (c) Tobacco use (n) Alcohol use (n) Methamphetamine use (b) Diabetes (n) Previous cardiovascular disease (n) Chronic hypertension(b) If chronic hypertensive, number of medications (o) Anaemia (n) Neurological disease (n) Respiratory disease (n) Renal disease (n) Inflammatory bowel disease (b) Autoimmune disease (o) Depression (b) Heredity for pre-eclampsia (o) First degree relative with hypertensive disorders (b) Partner’s heredity for pre-eclampsia (b) Pre-pregnancy weight (c) Pre-pregnancy height (c) Antenatal care (n) Gestation at first presentation for antenatal care (c) Systolic blood pressure at booking (c) Diastolic blood pressure at booking (c) Proteinuria at booking (o) On hypertensive treatment at booking (b) Aspirin use in this pregnancy (b) Gestation aspirin was started (c) Calcium use in this pregnancy (b) Gestation calcium was started (c) Estimated due date (N/A) Estimated due date calculated by (o)	Cord blood taken (b) Neonatal outcome (n) **Placental outcomes** Placental sample taken (b) Date for placental sample (N/A) Placental weight (c) **Daily hospital forms** Days since admitted (n) Highest systolic blood pressure (c) Highest diastolic blood pressure (c) Lowest oxygen saturation (c) Highest respiratory rate (c) Intubated (b) CPAP (b) Antihypertensive medication (n) Magnesium sulphate (b) Nitroglycerin/Tridil (b) Diuretic (b) Special investigations performed (n) Seen by other specialities (n) Haemoglobin (c) Platelets (c) Urea (c) Creatinine (c) Arterial pH (c) Arterial pO2 (c) Arterial pCO2 (c) Arterial lactate (c) Magnesium (c) Blood taken for freezing (b) **Transcranial doppler** Date and time for examination (N/A) Oxygen saturation when start of measurement (c) Pulse at start of examination (c) Systolic blood pressure at start of examination (c) Diastolic blood pressure at start of examination (85) Oxygen (n) End-tidal CO_2_ (c) Magnesium sulphate at examination (b) Magnesium sulphate before examination (b) If yes, how many hours since finished (o) Nitroglycerin/Tridil at examination (b) Antihypertensive medication at examination (o)	*HELLP syndrome* *Liver enzymes > 500 IU/L* *Liver rupture or hematoma* *Admitted to ICU* *Maternal death* *PPH* *Laparotomy for any reason apart from caesarean section* *Glasgow Coma Score < 13* *Myocardial infarction* *Intubation* *Sepsis* *Coma* *Venous thromboembolism* *Intrauterine fetal death* *Perinatal or infant mortality* *Abruptio placenta* *Mild to moderate pre-eclampsia without complications* *None (pregnant control)* **Cognitive function testing** MoCa (scanned) CFQ (c) **Long-term outcome 3-12 months** Date (N/A) Neonatal death within 6 weeks after expected due date (b) Neonatal death at any point after discharge (b) Maternal systolic blood pressure (c) Maternal diastolic blood pressure (c) Is mother on blood pressure treatment (b) Final hypertensive diagnosis (n) Any persisting neurological symptoms (b) Glasgow Outcome Scale (o)
**Symptoms before inclusion** Edema (n) Visual disturbances (n) Time of onset for visual disturbances (n) Headache (n) Headache onset (n) Epigastric/abdominal pain (n) Tightness in the chest (b) Shortness of breath (b) Focal neurological deficits (n) Tendon reflexes (n) Nausea (b) Vomit (b) Confusion (b) Twitching or jerking arms or legs (b) People around her noticing that she was absent in mind (b) Difficulty concentrating (b) Speech affected (b) Hearing affected (b) Mood changes (n) Anxiety (b) Feel like end of the world was coming (b) Dizziness (b) Weakness/paralysis (b) Other symptoms not listed (free text) Highest blood pressure before inclusion /fit (c) Highest diastolic blood pressure before inclusion /fit (c) Blood pressure recorded before/after fit (n) Date of convulsion if applicable (N/A) Fitted postpartum (b) Proteinuria (o) Lowest platelet count (c) Highest AST (c) Lowest haemoglobin (c) Highest creatinine (c) Blood pressure medication before the event/inclusion (n) Magnesium sulphate before the event/inclusion (b) If eclamptic fit, where did it occur (n)	GCS score at examination (c) Neurological deficits at start of examination (text) Depth left side (c) Systolic velocity left side (c) Mean velocity left side (c) Diastolic velocity left side (c) Cerebral perfusion pressure left side (c) Autoregulatory index left side (c) Depth right side (c) Systolic velocity right side (c) Mean velocity right side (c) Diastolic velocity right side (c) Cerebral perfusion pressure right side (c) Autoregulatory index right side (c) **Brain MRI** MRI performed (b) Date of MRI (N/A) *MR data saved in the electronic radiology system for later entry* **Echocardiography and MRI heart** Diagnosis of pulmonary edema in relation to delivery (n) Date of echocardiography (N/A) Date of cardiac MRI (N/A) Inclusion criteria (o) If pulmonary oedema, how was diagnosis set (o) Blood sample (b) LV diameter in diastole (c) LV end-systolic diameter (c) Interventricular septal thickness in diastole (c) Posterior wall thickness in diastole (c) LV mass (c) LV mass index (c) Relative wall thickness (c) Left ventricular ejection fraction (by Simpson’s biplane) (c) Fractional shortening (c) Tissue doppler SA wave velocity (c) Myocardial performance index (c) Global longitudinal strain (c) Regional wall motion abnormalities (c) E wave height (c) A wave height (c) E:A ratio (c) E wave deceleration time (c) Isovolumetric relaxation time (c) E’ septal (c) E’ lateral (c) E’average (c) E:E’septal (c) E:E’lateral (c) E:E’average (c) Pulmonary vein pulse wave doppler (c) Left atrial diameter (c) Left atrial area (c) Left atrial volume (c) Left atrial volume indexed to BSA (c)	

APGAR; appearance, pulse, grimace, activity, respiration, AST; aspartate transaminase, CFQ; Cognitive Failure Questionnaire, CPAP; continuous positive airway pressure, GCS; Glasgow Coma Scale, HELLP; haemolysis, elevated liver enzymes, low platelets, HIV; human immune deficiency virus, ICU; intensive care unit, LV; left ventricle, MoCA; Montreal Cognitive Assessment, MRI; magnetic resonance imaging, PPH; postpartum haemorrhage, PRES; posterior reversible encephalopathy syndrome. Nominal (n); binary (b); ordinal (o); continuous (c).

**Table 2 cells-10-00959-t002:** Biological samples collected for the PROVE biobank.

Type of Sample	Sampling	Volume	Tubes, Processing
Blood	At inclusion, at delivery, during hospital stay	Maximum of 12 mL/sample	EDTA plasma, spinned, 1 mL aliquots
Placenta: foetal surface	At delivery	RNAlater, 4 × 1 cm^3^	Removal of RNA later, 2 mL aliquots
Placenta: foetal surface	At delivery	Frozen sections 1 cm^3^	2 mL aliquots
Placenta: maternal surface	At delivery	RNAlater, 4 × 1 cm^3^	Removal of RNA later, 2 mL aliquots
Placenta: maternal surface	At delivery	Frozen sections 1 cm^3^	2 mL aliquots
Cord Blood (mixed venous–arterial)	At delivery	Maximum 12 mL	EDTA plasma, spinned, 1 mL aliquots
Urine	At inclusion	Maximum 12 mL	Sterile tube, spinned, 1 mL aliquots
Cerebrospinal fluid	At delivery	Maximum 2 mL	Sterile tube, spinned, 1 mL aliquots
Saliva	At inclusion	Stored in a 3 cc cup	2 mL aliquots

EDTA; ethylenediaminetetraacetic acid, RNA; ribonucleic acid.

**Table 3 cells-10-00959-t003:** Background characteristics of the women included in the biobank until March 2020, by complication at inclusion.

	Neurology *	Pulmonary Oedema	HELLP/Renal Impairment	Pre-Eclampsia **	Normotensive
***n***	86	43	23	54	38
**AT BASELINE**					
**Maternal age (years)**	22.8 (6.1)	28.8 (6.9)	29.2 (6.8)	26.1 (5.8)	29.1 (6.3)
**Nulliparous n (%)**	60 (70)	18 (42)	7 (30)	28 (52)	9 (24)
**BMI (kg/m^2^)**	26.4 (8.1)	32.2 (9.2)	29.8 (6.4)	29.1 (7.3)	27.6 (6.8)
Missing ***	16	5	7	3	4
**HIV n (%)**	10 (12)	11 (26)	6 (26)	8 (15)	7 (18)
**Chronic hypertension n (%)**	7 (8)	2 (5)	5 (22)	9 (17)	0 (0)
**Gestation at first presentation for antenatal care (weeks)**	17.4 (8)	15.3 (7.6)	18.0 (6.9)	16.0 (7.4)	16.6 (7.7)
**AFTER INCLUSION**					
**Gestation at delivery (weeks)**	33.4 (4.3)	31.8 (5.0)	30.6 (5.0)	34.0 (4.2)	36.1 (3.9)
**Mode of delivery n (%)**					
Vaginal	25 (29)	11 (26)	8 (35)	15 (28)	8 (21)
Planned CS	1 (1)	2 (5)	0 (0)	5 (9)	24 (63)
Emergency CS	60 (70)	30 (70)	15 (65)	34 (63)	6 (16)
**Birthweight (grams)**	2093 (906)	1748 (969)	1315 (588)	2078 (961)	2761 (877)
**OUTCOMES n (%)**					
**Severe hypertension**	35 (41)	29 (67)	16 (70)	14 (26)	0 (0)
**Eclampsia**	82 (95)	0 (0)	0 (0)	0 (0)	0 (0)
**Recurrent eclampsia**	26 (30)	0 (0)	0 (0)	0 (0)	0 (0)
**GCS < 13**	22 (26)	0 (0)	0 (0)	0 (0)	0 (0)
**Stroke**	3 (4)	0 (0)	0 (0)	0 (0)	0 (0)
**Blindness**	1 (1)	0 (0)	0 (0)	0 (0)	0 (0)
**Pulmonary oedema**	5 (6)	43 (100)	3 (13)	0 (0)	0 (0)
**HELLP**	19 (22)	7 (16)	22 (96)	1 (2)	0 (0)
**Renal impairment**	16 (19)	5 (12)	9 (39)	1 (2)	0 (0)
**Admitted to ICU**	10 (12)	6 (14)	1 (4)	0 (0)	0 (0)
**Postpartum haemorrhage**	9 (11)	3 (7)	2 (13)	3 (5)	2 (4)
**Intubation**	15 (17)	6 (14)	1 (4)	0 (0)	0 (0)
**Intrauterine foetal death**	12 (14)	8 (19)	7 (30)	5 (9)	0 (0)
**Venous thromboembolism**	2 (2)	1 (2)	0 (0)	0 (0)	0 (0)
**Abruptio placentae**	5 (6)	2 (5)	2 (9)	2 (4)	1 (3)
**GOS at two months**					
Mild or no disability	63 (96)	33 (100)	10 (100)	42 (100)	26 (100)
Severe disability	1 (1)	0 (0)	0 (0)	0 (0)	0 (0)
Death	2 (3)	0 (0)	0 (0)	0 (0)	0 (0)
Missing	20	10	13	12	12

* Eclampsia, stroke, or blindness. ** Pre-eclampsia with or without acute severe hypertension, without organ complications. *** Missing values for BMI due to missing antenatal charts or antenatal charts without registration of BMI.; BMI, body mass index; GCS, Glasgow Coma Scale, GOS, Glasgow Outcome Scale, HELLP, haemolysis, elevated liver enzymes, and low platelets; HIV, human immunodeficiency virus; ICU, intensive care unit; continuous variables are presented as means with standard deviations. Categorical variables are presented as numbers with percentages.

## Data Availability

The datasets used and/or analysed during the current study are available from the corresponding author on reasonable request.

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
