# Peer review of "PROVE—Pre-Eclampsia Obstetric Adverse Events: Establishment of a Biobank and Database for Pre-Eclampsia"

_cells, 2021, doi:10.3390/cells10040959_

Round 1

Reviewer 1 Report

Congratulations to the authors for setting up the protocol and in this way they opened up opportunities for research on very concretely specified biological material.

Cord blood is also described among the biological material. It is not specified whether arterial, venous or mixed cord blood is taken (lines 120 and 121 and Table 2).

Table 2 also does not specify a serum as a matrix (?) 

Author Response

We than the reviewer for the favorable comments on our manuscript. 

We have clarified in the manuscript that cord blood is mixed venous-arterial. We only obtained plasma samples due to difficulties in handling serum.

Reviewer 2 Report

Thank you very much for allowing me tome to review manuscript entitled: “PROVE –Pre-eclampsia Obstetric Adverse Effects Establishment of a biobank and database for pre-eclampsia” submitted to "Cells". The purpose of this article was to improving understanding of pathophysiology, prediction and potentially future drug discovery by collectin biological specimens, detailed clinical data, tests and biophysical examinations including MRI of the brain, MRI of the heart, transcranial Doppler, echocardiography and establishing a pre -eclampsia biobank in South Africa.
Authors raise a very important topic and relevant issue. Despite the advances in medicine and many studies, the etiology of preeclampsia still remains not fully understood. Therefore, another voice in the discussion about etiopathogensis of this specific for human pregnancy complication such as preeclampsia is very important.
The research is very original and important. this paper takes a interesting approach to the topic of preeclampsia, a very serious complication of pregnancy.
It should be emphasized that the research team at Tygerberg Hospital also collaborated with leading academic centers in Australia, Sweden, the Netherlands, UK and USA.
The methodology of work together with the inclusion and exclusion criteria have been described in detail. The study has been registered and yearly is updated to the local Health Research Ethics Committee.
This biobank is ongoing project. To this days 244 women have been included into the biobank. It needs to be highlighted that will continue after the COVID-19 pandemia.
Therefore, it seems justified to indicate that this is a preliminary study that this research and biobanking will be continued. 
It is a well done study. Citations and reference range are correctly selected.
  I recommend this manuscript for publication.

Author Response

We thank the reviewer for the favorable comments.

We have re-inforced that the biobank is an ongoing project in lines 247-248 which now reads;

"and will continue to include women according to the research questions and sample size described above."